# A Growth Factor-Free Co-Culture System of Osteoblasts and Peripheral Blood Mononuclear Cells for the Evaluation of the Osteogenesis Potential of Melt-Electrowritten Polycaprolactone Scaffolds

**DOI:** 10.3390/ijms20051068

**Published:** 2019-03-01

**Authors:** Andreas Hammerl, Carlos E. Diaz Cano, Elena M. De-Juan-Pardo, Martijn van Griensven, Patrina S.P. Poh

**Affiliations:** 1Experimental Trauma Surgery, Klinikum rechts der Isar, Technical University of Munich, 81675 Munich, Germany; hammerl.andi@gmx.de (A.H.); cadiazcano@gmail.com (C.E.D.C.); martijn.vangriensven@tum.de (M.v.G.); 2Institute of Technology Costa Rica, Department of Biotechnology Engineering, Cartago 30101, Costa Rica; 3Institute of Health and Biomedical Innovation, Queensland University of Technology (QUT), 4059 Brisbane, Australia; elena.juanpardo@qut.edu.au; 4Julius Wolff Institut, Charité—Universitätsmedizin Berlin, 13353 Berlin, Germany

**Keywords:** melt electrowriting, bone tissue engineering, calcium phosphate, human primary cells

## Abstract

Scaffolds made of biodegradable biomaterials are widely used to guide bone regeneration. Commonly, in vitro assessment of scaffolds’ osteogenesis potential has been performed predominantly in monoculture settings. Hence, this study evaluated the potential of an unstimulated, growth factor-free co-culture system comprised of osteoblasts (OB) and peripheral blood mononuclear cells (PBMC) over monoculture of OB as an in vitro platform for screening of bone regeneration potential of scaffolds. Particularly, this study focuses on the osteogenic differentiation and mineralized matrix formation aspects of cells. The study was performed using scaffolds fabricated by means of a melt electrowriting (MEW) technique made of medical-grade polycaprolactone (PCL), with or without a surface coating of calcium phosphate (CaP). Qualitative results, i.e., cell morphology by fluorescence imaging and matrix mineralization by von Kossa staining, indicated the differences in cell behaviours in response to scaffolds’ biomaterial. However, no obvious differences were noted between OB and OB+PBMC groups. Hence, quantitative investigation, i.e., alkaline phosphatase (ALP), tartrate-resistant acid phosphatase (TRAP) activities, and gene expression were quantitatively evaluated by reverse transcription-polymerase chain reaction (RT-qPCR), were evaluated only of PCL/CaP scaffolds cultured with OB+PBMC, while PCL/CaP scaffolds cultured with OB or PBMC acted as a control. Although this study showed no differences in terms of osteogenic differentiation and ECM mineralization, preliminary qualitative results indicate an obvious difference in the cell/non-mineralized ECM density between scaffolds cultured with OB or OB+PBMC that could be worth further investigation. Collectively, the unstimulated, growth factor-free co-culture (OB+PBMC) system presented in this study could be beneficial for the pre-screening of scaffolds’ in vitro bone regeneration potential prior to validation in vivo.

## 1. Introduction

Tissue engineering and regenerative medicine (TERM) show great potential in providing personalized treatment to improve bone regeneration. Over the years, biochemical strategies (i.e., growth factors, cell therapy, gene therapy etc.), biophysical strategies (i.e., scaffolds), and the combinations of both have been widely explored—all showing great potential. With the recent development in the European Union Medical Device Regulation (EU-MDR), the translational potential of biochemical strategies in an orthopaedic setting has been increasingly challenging. In recent years, the use of scaffolds as temporary tissue expanders to create an initial void space to allow for tissue-in growth has been extensively studied [1] and proven effective in numerous in vivo studies [2,3].

Fabrication of bone scaffolds can be achieved through conventional or additive manufacturing (AM) techniques. Conventional techniques used for the manufacturing of scaffolds intended for bone tissue engineering, e.g., solvent casting, gas foaming, freeze-drying, or solution electrospinning, often require organic solvents for processing. Moreover, the manual processes are not easily scalable to a highly reproducible manner [4]. On the other hand, AM techniques permit for precise control over the scaffold architecture in a highly reproducible manner [5]. Melt electrowriting (MEW), one of the recent approaches in AM [6], offers an accurate layering of highly uniform micrometric fibrous structures using molten thermoplastic, mitigating the need for organic solvents [7]. Among the different thermoplastics, polycaprolactone (PCL) has been widely used in MEW for scaffolds intended for bone regeneration [4,7,8,9]. Besides its absorbability and biocompatibility [10], PCL has good rheological and viscoelastic properties compared to other synthetic polymers [11]. Additionally, PCL’s low melting temperature (60 °C), semi-crystallinity with rapid solidification, and thermal stability make it a good candidate for MEW processing [7].

Although PCL scaffolds have demonstrated capabilities in supporting cell adhesion and proliferation after surface modification with plasma treatment or sodium hydroxide (NaOH) etching, they lack osteogenic capability [12]. To circumvent this, PCL scaffolds can be surface-coated with calcium-phosphate (CaP)-based apatite, providing the favourable biological potential to the scaffold [13], especially the capability for osteogenesis [12,14]. Commonly, in vitro examinations for the potential for scaffolds to facilitate bone formation have been performed in monoculture settings, e.g., with either osteoblast (OB) or mesenchymal cell [12,14,15,16], which poorly recapitulate the physiological microenvironment of bone regeneration. Consequently, this led to a low clinical translation rate of absorbable scaffolds for medical applications [17], most likely due to the unexpected inefficacy of the scaffolds to promote bone regeneration in vivo. Hence, there is a need for improved in vitro culture system for pre-screening of scaffolds’ bone regeneration potential prior to further performance validation in vivo, which are associated with higher cost and ethical considerations.

Inevitably, when utilizing absorbable scaffolds as a temporary supporting structure to guide tissue regeneration, circulating peripheral blood mononuclear cells (PBMC) will infiltrate into the regenerating niche and participate in tissue regeneration [18,19,20,21], as well as a degradation of scaffolds [22]. Hence, the aim of this study was to evaluate and directly compare the effectiveness of OB-only monoculture and OB+PBMC co-culture in an unstimulated, growth factor-free culture system for the in vitro evaluation of bone regeneration potential for MEW PCL scaffolds surface-coated with CaP (PCL/CaP), with a focus on the osteogenic differentiation and mineralized matrix formation aspects.

## 2. Results

### 2.1. Scaffolds’ Morphology

Through the MEW technique, medical-grade PCL scaffolds with a pore size of around 100 μm and a microfilament diameter between 10 and 13 μm were produced and characterized via scanning electron microscopy (SEM) (Figure 1a). After CaP post-processing, CaP particles were found on the scaffolds’ surface, as shown by SEM in Figure 1b.

### 2.2. Visualization of Cell Morphology 

As shown in the confocal images of Figure 2, both OB and PBMC were attached to the scaffold’s microfilaments. For the PCL/CaP scaffolds cultured with PBMC only, multi-nucleated giant cells formation was observed on Day 35 and became more apparent on Day 63. On the PCL scaffolds cultured with PBMC only and all scaffolds co-cultured with OB+PBMC, no multi-nucleated giant cells were observed. Nonetheless, for PCL and PCL/CaP scaffolds cultured with OB+PBMC or OB-only, cells continued to proliferate from Day 35 to Day 63, as indicated qualitatively by the increasing amount of cell nuclei and actin filaments.

### 2.3. Visualization of the Mineralized Extracellular Matrix

The extent of extracellular matrix (ECM) mineralization on scaffolds cultured with OB-only or in co-culture with PBMC was assessed by means of von Kossa staining. Digital microscopy images shown in Figure 3 revealed an increase in the ECM density over time for all groups. It is noteworthy to highlight that, on Day 7, PCL/CaP scaffolds cultured with OB+PBMC showed the highest density of ECM compared to all other groups. However, no mineralized ECM was formed on Day 7. On Days 35 and 63, qualitatively, PCL/CaP scaffolds cultured with OB-only or OB+PMBC showed similar levels of mineralized ECM formation, as indicated by the black patches (von Kossa staining) found on the cell-scaffold constructs. On the contrary, no mineralized matrix was observed on PCL scaffolds cultured with OB-only or OB+PMBC.

### 2.4. ALP and TRAP Activity

ALP activity was assessed to determine the extent of osteogenic potential of PCL/CaP scaffolds when co-cultured with OB+PBMC as compared to OB mono-culture. As illustrated in Figure 4a, ALP activity was similar in both groups throughout the experimental time points.

On the other hand, TRAP activity was evaluated as an indicator for PMBC’s osteoclastic potential when cultured on PCL/CaP scaffolds, with or without the presence of OB. In the presence of OB, TRAP activity was detected on Day 14, but diminished thereafter. Conversely, in the PBMC monoculture group, high TRAP activity was detected on Days 14 and 35, and diminished on Day 63. Overall, the TRAP activity in the OB+PBMC co-culture group remained significantly lower compared to that of the PBMC monoculture group for all experimental time points (Figure 4b).

### 2.5. Analysis of Gene Expression

Scaffolds provide a much larger surface area for cell growth compared to 2D culture, such that the time progression of cell growth to cell differentiation to matrix mineralization on scaffolds will be extended [13]. In addition, cells of different status, i.e., differentiated cells on scaffolds’ surfaces, proliferating cells across the pores of scaffolds, etc., will concurrently present across the entire scaffold construct. It is widely known that osteogenic differentiation occurs in a highly confluence environment [23]. Hence, based on the light microscopy image observation (Figure 3), mRNA expression was evaluated in the later phase of culture (i.e., 14, 35, and 63 days)—~80% confluence cells on scaffolds with OB or OB+PBMC.

Osteogenic-related genes (Figure 5), namely, RunX2, ALP, and OCN, were assessed for scaffolds cultured with OB alone or in co-culture with PBMC. Overall, both OB and OB+PBMC culture groups showed a temporal expression of RunX2, ALP, and OCN genes over the experimental period. However, no significant differences in the expression levels of these genes were observed between the culture groups. As expected, higher expression of the ColIAI gene (Figure 5), an indicator of the formation and secretion of ECM, was observed in OB-only and OB+PBMC culture groups compared to the PBMC-only group. However, the ColIAI expression level was similar in both OB-only and OB+PBMC culture groups.

On the other hand, osteoclastic related genes (Figure 5), namely, TRAP and CTSK, were evaluated for scaffolds cultured with PBMC, with or without OB. TRAP gene expression, an intermediate marker of osteoclastogenesis, was more prominent in the PBMC-only group compared to the OB+PBMC group. On the contrary, CTSK gene expression, a late marker for osteoclastogenesis, was highly expressed in both PBMC-only and OB+PBMC co-cultures, with no significant difference observed between the groups over the experimental time points.

RANKL is known to be secreted by OB and binds to the RANK receptor on the osteoclast precursor, regulating the formation of multinucleated osteoclasts. On the other hand, osteoprotegerin (OPG) is known to be secreted by OB as a decoy receptor for RANKL to inhibit osteoclast maturation through RANK–RANKL interactions [24]. Monoculture of OB showed a marginal expression level of RANKL for all experimental time points (Figure 5). Conversely, co-culture of OB+PBMC showed significantly higher levels of RANKL gene expression on Day 14 compared to that of the OB monoculture. Thereafter, the RANKL gene expression level was lowered with no significant differences between the OB and OB+PBMC groups. On the other hand, OB+PBMC groups showed marginal expression of RANK at all time points, while the PBMC mono-culture group showed increased levels of RANK expression over time. At Day 63, the PMBC group showed a significant higher RANK expression level compared to that of the OB+PBMC co-culture group. In terms of OPG, both OB and OB+PBMC groups showed an increased level from Day 14 to Day 35. However, no significant difference was observed between groups.

## 3. Discussion

This present study investigated and compared the effectiveness of OB-only monoculture vs. OB+PBMC co-culture in an unstimulated, growth factor-free culture system for the in vitro evaluation of scaffolds’ bone regeneration potential, with a focus on the osteogenic differentiation and mineralized matrix formation aspects. Through the initial qualitative results, i.e., cell morphology (Figure 2) and matrix mineralization (Figure 3), it was apparent that the scaffolds’ biomaterial composition can greatly influence cell behaviours [25]. Large multi-nucleated cells can be easily spotted on PCL/CaP scaffolds, but none was found on PCL scaffolds (Figure 2). Hence, further investigation for the potential of OB+PBMC co-culture as a system thatbetter mimics the physiological microenvironment of the regenerating bone niche was only evaluated using PCL/CaP scaffolds, with OB and PBMC-only monoculture as control groups.

Once adhered to the biomaterials, OB begins the proliferation and maturation process, initiating the production of proteins related to mineralization [26] leading to the formation of the mineralized matrix. The von Kossa stained images, the ALP activity assay, and the osteogenic-related gene expression (i.e., RunX2, ALP, OCN, and ColIAI) data showed that PCL/CaP scaffolds caused similar levels of osteogenesis when cultured with OB or OB+PBMC. With or without the presence of PBMC, PCL/CaP scaffolds were able to support OB growth and the secretion of mineralized ECM in the absence of exogenous osteogenic factors. Commonly, a combination of osteogenic differentiation supplements, namely, dexamethasone, L-ascorbic acid, and β-glycerophosphate, is usually added to the medium to improve OB cell differentiation and the synthesis of matrix proteins [26]. In this study, using an unstimulated, growth factor-free culture system, it was clearly demonstrated in Figure 3 that a mineralized matrix was observed on PCL/CaP but not PCL scaffolds (Figure 3), regardless of whether scaffolds were cultured with OB or OB+PBMC. This clearly indicated that, instead of the presence of PBMC, the CaP coating on the PCL scaffolds’ surfaces is the predominant driving force for mineralized matrix formation. The potential of CaP-coating on scaffolds’ surfaces to induce mineralized ECM formation has been previously shown by Poh et al. [13] and Vaquette et al. [12] using ovine OB. Consequently, within the context of osteogenic differentiation and mineralized matrix formation, there seem to be no apparent benefits of using OB+PBMC co-culture over the OB monoculture system to evaluate the bone regeneration potential of scaffolds.

From the preliminary result (Figure 3) of this study, it seems like the cell proliferation or ECM secretion mechanism of OB was facilitated in the presence of PBMC. At Day 7, all scaffolds (PCL and PCL/CaP scaffolds) cultured with OB+PBMCs showed more ECM across the entire scaffold volume compared to the OB-only group. This indicated that the presence of PBMC promoted the secretion of ECM by OB in the early phase. This phenomenon has been previously reported, whereby PBMC were capable of stimulating OB proliferation presumably through the cytokines secreted by PMBC [27,28]. The elucidation of the interaction of OB and PBMC in relation to the cells proliferative or non-mineralized ECM secretion is beyond the scope of this study but would be of interest for subsequent study.

Another noteworthy observation in this study was that, in the PBMC-only culture, multi-nucleated cells were formed on PCL/CaP but not on PCL scaffold surfaces (Figure 2). However, the presence of OB seems to have suppressed the capability of PMBC to form multi-nucleated cells on PCL/CaP scaffolds. This was in line with the observation of high TRAP activity (Figure 4b) and gene expression (Figure 5) in the PBMC-only culture group compared to that of OB+PBMC groups. Surprisingly, the CTSK gene was highly expressed in PCL/CaP scaffolds cultured with OB+PBMC and PBMC. It is beyond the scope of this study to unravel the underlying mechanism for the upregulation of CTSK gene expression. It may be possible that the expression of CTSK in the OB+PBMC culture system is associated with the upregulation of osteoclastogenesis factor, RANKL, as previously reported by Fujisaki et al. [29]. It may also be possible that, although CTSK is generally known as the late-osteoclastic marker, that CTSK is conserved during the fusion of mononuclear cells, as well as during macrophage activation and differentiation [30]. The polarization of the PBMC in the co-culture system may have contributed to the accelerated OB proliferation or non-mineralized ECM secretion. This could be a potential direction of study for future studies.

Collectively, this study showed that it could be noteworthy to utilize a co-culture system, i.e., OB+PBMC for the screening of scaffolds’ bone regeneration potential. Although this study showed no differences in terms of osteogenic differentiation and ECM mineralization, preliminary qualitative results indicated an obvious difference in the cell/non-mineralized ECM density between scaffolds cultured with OB or OB+PBMC. Through this study, a cost-effective in vitro platform, i.e., an unstimulated, growth factor-free co-culture (OB+PBMC) system, is presented as a potential platform for the evaluation of scaffolds intended for bone regeneration application—a system that better mimics the physiological microenvironment of the regenerating bone niche.

## 4. Materials and Methods

### 4.1. Scaffolds Fabrication and Post-Processing

All scaffolds were fabricated by MEW. Briefly, molten medical-grade PCL (Purasorb PC-12, Corbion Purac, AC Gorinchem, The Netherlands) was extruded at 90 °C with an applied voltage of 11 kV and air pressure of 1.16 bar. The scaffolds were extruded as a 12 × 12 cm square sheet onto a dust-free grounded metal stage. Scaffolds comprised of a total of 10 layers of PCL microfilaments, angled at 90° between each layer with a pore size of approximately 100 µm. Subsequently, the scaffolds were cut into 8 × 8 mm square sheet using a laser cutter.

To improve the osteogenic potential of scaffolds, scaffolds were coated with a thin layer of CaP (PCL/CaP). Briefly, scaffolds were immersed in 70% ethanol for 30 min under vacuum and treated with pre-warmed 1 M NaOH for 40 min at 37 °C. Then, scaffolds were rinsed with distilled water (dH_2_O) and covered with 10× simulated body fluid (SBF) of pH 7.4 for 30 min [12], with a change of fresh 10× SBF after 30 min. Finally, the scaffolds were subjected to 0.5 M NaOH treatment for 5 min and gently washed with dH_2_O until the wash solutions reached pH 7.

For initial assessment for cell morphology and matrix mineralization (von Kossa), PCL and PCL/CaP scaffolds were used. Due to the apparent effect of the biomaterial of scaffolds on cell behaviour, especially PBMC (Figure 2) and matrix mineralization (Figure 3), only PCL/CaP scaffolds were selected for the subsequent experiments. This is to reduce experimental variables and enable direct comparison of the effectiveness of OB monoculture vs. OB+PBMC co-culture system for the in vitro evaluation of scaffolds’ bone regeneration potential.

### 4.2. Cell Isolation

OB: Cells were isolated from the femur head or shoulder during implantation of endoprosthesis of otherwise healthy donors. All donors gave their informed consent. The study was approved by the institutional review board and carried out following the declaration of Helsinki principles. Briefly, the cancellous bone was cracked into small pieces ranging from 1 to 4 mm, washed twice with a phosphate buffer solution (PBS), and transferred into a 175 cm^2^ tissue culture flask. Cells were cultured in growth medium made of low-glucose DMEM media supplemented with 10% fetal bovine serum (FBS), a 1% penicillin and streptomycin solution, and 0.2 M L-ascorbate-2-phosphate. Cells were maintained at 37 °C and 5% CO_2_ in a humidified incubator. Cells of Passage 3 were used for all experiments.

PBMC: Cells isolations were performed by Ficoll density gradient centrifugation of blood obtained from healthy donors. Briefly, 30 mL of blood was carefully laid on top of 20 mL of lymphocyte separation media (LSM) 1077 (PAA laboratories) and centrifuged at 1000× *g* at 22 °C for 20 min with disengaged brakes. After this, the cell layer at the interphase of plasma and the LSM was carefully aspirated and transferred into a new Falcon tube. Cells were washed with PBS and centrifuged at 650× *g* at 22 °C for 10 min with brakes engaged (repeat twice). Finally, cells were re-suspended in the growth medium.

### 4.3. Scaffold Sterilization and Cell Seeding

Scaffold sterilization was performed by immersion in 70% ethanol for 30 min and drying. Excess 70% ethanol was gently aspirated to avoid stacking of scaffolds and left to air-dry inside a sterile biosafety cabinet. Prior to cell seeding, scaffolds were treated with 20 min of ultraviolet (UV) light on each side. Each scaffold was seeded with 8.0 × 10^4^ OB suspended in 20 µL of growth medium, left for an hour at 37 °C and 5% CO_2_ in a humidified incubator to allow cell adhesion and fed with 700 µL of growth medium. Medium change was performed every three days. One week later, scaffolds were transferred to a new 24-well plate and seeded with 5 × 10^5^ freshly isolated PBMCs suspended in 20 µL of growth medium. Scaffolds seeded with only OB or PBMC were used as experimental control groups. Every three days, 350 µL of media was substituted with fresh growth medium.

### 4.4. Visualization of Cell Morphology

Cell morphology was visualized using confocal laser scanning microscopy (CLSM) (Olympus Fluoview FV10i, Olympus Europa Holding GmbH, Hamburg, Germany) on samples, i.e., PCL and PCL/CaP scaffolds cultured with OB, PBMC, or OB+PBMC, stained with Hoechst and Phalloidin. Briefly, cells were washed with PBS, fixed with 3.7% formaldehyde for 20 min, treated with a 0.2% Triton X-100 solution for 5 min, and washed with PBS. Then, samples were incubated with the Phalloidin (0.05 µg/mL) solution in PBS for 60 min in the dark. After 45 min, the Hoechst (0.5 µg/mL) solution was added. Finally, samples were washed twice with PBS and kept in PBS at 4 °C until CLSM imaging. The assay was performed in duplicate.

### 4.5. Visualization of Mineralized Cell Matrix by Von Kossa Staining

Cell-scaffold constructs, i.e., PCL and PCL/CaP scaffolds cultured with OB or OB+PBMC, were harvested, washed twice with PBS, and fixed with ice-cold methanol for 15 min. Cell-scaffold constructs were then incubated sequentially in the following solutions at room temperature: (i) a 3% silver nitrate solution for 30 min, (ii) a 1% pyrogallol solution for 3 min, (iii) 5% sodium thiosulphate for 5 min, (iv) a Kernechtred solution for 5 min, and (v) 96% ethanol for 1 min. In between the incubation steps, cell–scaffold constructs were washed with distilled water at least three times. Stained samples were kept in PBS at 4 °C and imaged with a digital microscope (VHX-1000, Keyence, Neu-Isenburg, Germany). The assay was performed in duplicate.

### 4.6. Cell Proliferation

A Quant-IT^TM^ PicoGreen^®^ ds-DNA assay kit (Invitrogen- Thermo Fischer Scientific GmbH, Berlin, Germany) was used for the quantification of cells. At Days 2, 7, 14, 35, and 63, samples, i.e., PCL/CaP scaffolds cultured with OB, PBMC, or OB+PBMC, were transferred into a new 48-well plate, washed with PBS, and topped with 400 µL of a Tris-EDTA (1× TE) solution (10 mM Tris-HCl, 1 mM EDTA, pH 7.5). Cells were lysed through three times of freeze-thaw cycles and mechanical abraded with a sterilized pipette tip. The cell lysate was transferred into a new tube and centrifuged at 1000× *g* for 10 min at 4 °C. The measurement of dsDNA was performed following the manufacturer instructions. Briefly, lambda DNA standards were prepared by eight-point serial dilution of a 2 µg/mL lambda DNA working solution (prepared by 1:50 dilution of a lambda DNA stock solution of 100 µg/mL in a 1× TE buffer). Afterwards, 100 µL of a sample or standard solution was transferred into each well of a 96-well plate in triplicate and 100 µL of a reaction solution (prepared by 1:200 dilutions of a Quant-iT PicoGreen reagent in a 1× TE buffer). Subsequently, the mixture was incubated for 5 min in the dark and measurement was taken using a Fluostar Omega Plate Reader measured with an excitation wavelength of 480 nm and an emission wavelength of 520 nm.

### 4.7. Alkaline Phosphatase (ALP) Assay

The ALP assay was performed for the measurement of cells’ ALP as an indication for early osteogenesis. Briefly, at Days 2, 7, 14, 35, and 63, samples, i.e., PCL/CaP scaffolds cultured with OB or OB+PBMC, were transferred to a new 48-well plate, washed twice with PBS, and incubated in 250 µL of an ALP reaction solution (3.5 mM 4-nitrophenyl phosphate di-sodium salt hexahydrate in an ALP buffer) for 30 min at 37 °C. A reaction solution without cells was used as a control. At the same time, an AP standard solution was prepared by serial dilutions of a 1 mM 4-nitrophenol solution in an ALP buffer (50 mM Glycine, 100 mM Tris-Base and 2 mM MgCl_2_) at pH 10.5. After incubation, 100 µL of supernatant diluted at a 1:1 ratio in PBS or a standard solution was transferred to a 96-well plate in triplicate. Finally, absorbance was measured at 405 nm using a Fluostar Omega Plate Reader. ALP measured was normalized with dsDNA content obtained from a PicoGreen assay.

### 4.8. Tartrate-Resistant Acid Phosphatase (TRAP) Assay

At Days 2, 7, 14, 35, and 63, the cell supernatant (50 µL) of the sample, i.e., PCL/CaP scaffolds cultured with PBMC or OB+PBMC, was transferred to a 96-well plate in triplicate and mixed with 150 µL of a reaction solution (5 mM 4-nitrophenyl phosphate di-sodium salt hexahydrate in a TRAP assay buffer) and incubated for 1 h at 37 °C. The reaction solution incubated with a fresh growth medium was used as a control. In parallel, a TRAP standard solution was prepared by serial dilution of a starting solution made of 450 μL of culture media, 1350 μL of a TRAP assay buffer (100 mM sodium acetate and 50 mM sodium tartrate dihydrate), and 200 μL of a 10 mM 4-nitrophenol solution. Afterwards, 200 µL of a serial standard solution was transferred to a 96-well plate in triplicate. Thereafter, 50 µL of 3 M NaOH was added to stop the reaction, and absorbance was measured at 405 nm using a Fluostar Omega Plate Reader. TRAP activity was normalized with dsDNA content obtained from a PicoGreen assay.

### 4.9. Gene Expression by Quantitative Reverse Transcription Polymerase Chain Reaction (qRT-PCR)

Three cell-scaffold constructs of the same experimental groups, i.e., PCL/CaP scaffolds cultured with OB, PBMC, or OB+PBMC, were pooled together, washed with PBS, and transferred to a tube containing 250 µL of a TRIZOL reagent and kept in −80 °C for at least 24 h. RNA isolation was performed by chloroform extraction. RNA quantification and quality control were performed with NanoDrop (Nanodrop Tech- Thermo Fischer Scientific GmbH, Berlin, Germany,). Reverse transcription to cDNA was performed in a C1000 Touch Thermal Cycler (Eppendorf, Hamburg, Germany) using a first-strand cDNA synthesis kit (Thermo Scientific, Waltham, MA, USA) following the manufacturer’s instructions. qPCR was performed in a CFX96 Real-Time System thermocycler using SsoFast EvaGreen Supermix (BioRad, Hercules, CA, USA) as a detection reagent. Gene expression was expressed as ΔΔ*C*T relative to the housekeeper (β-tubulin). The gene sequences used were listed in Table 1.

## Figures and Tables

**Figure 1 ijms-20-01068-f001:**
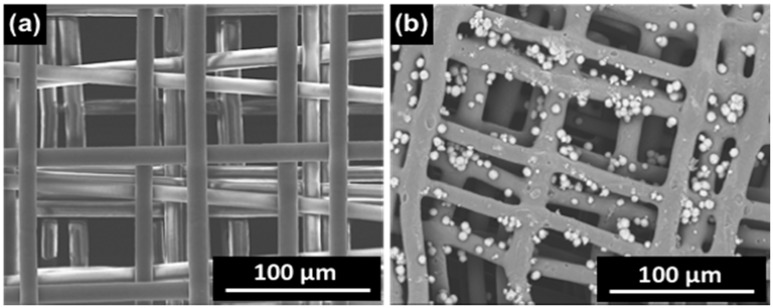
SEM images of melt electrowritten medical-grade polycaprolactone (PCL) scaffolds (**a**) as manufactured and (**b**) after coating with calcium phosphate (CaP) particles.

**Figure 2 ijms-20-01068-f002:**
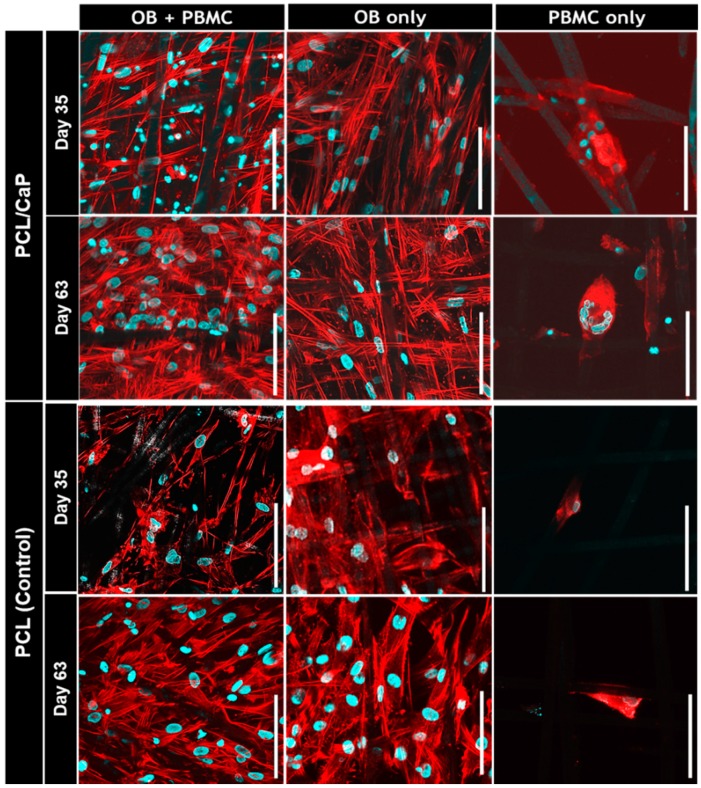
Cell-seeded polycaprolactone (PCL) with or without calcium phosphate (CaP) surface-coated scaffolds stained with Hoechst (blue—cell nuclei) and Phalloidin (red—actin filaments). OB = osteoblast; PBMC = peripheral blood mononuclear cell. Scale bar = 100 μm.

**Figure 3 ijms-20-01068-f003:**
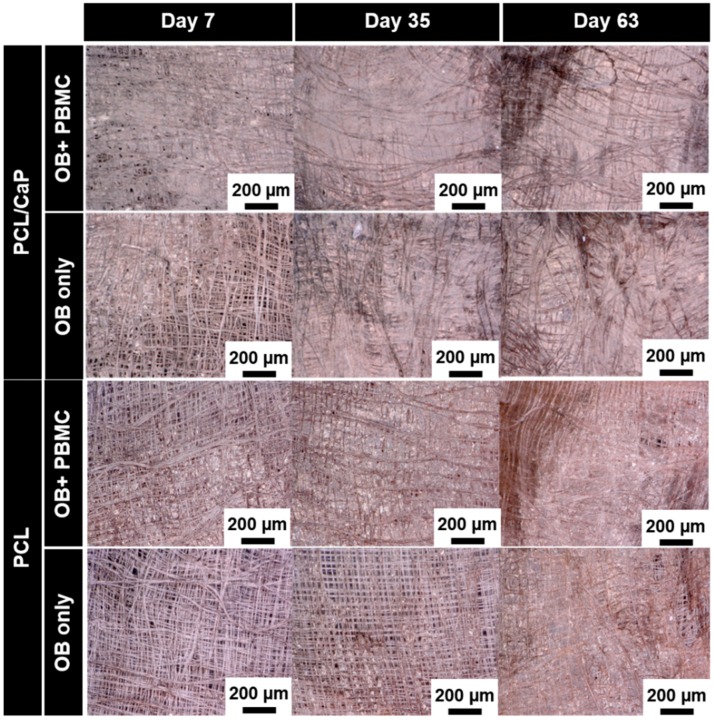
Digital microscopy images (100× magnification) of osteoblast-peripheral mononuclear cell co-cultures (OB+PBMC) and osteoblast monocultures (OB only) on polycaprolactone (PCL) scaffolds with or without calcium phosphate surface coating stained with von Kossa at different time points.

**Figure 4 ijms-20-01068-f004:**
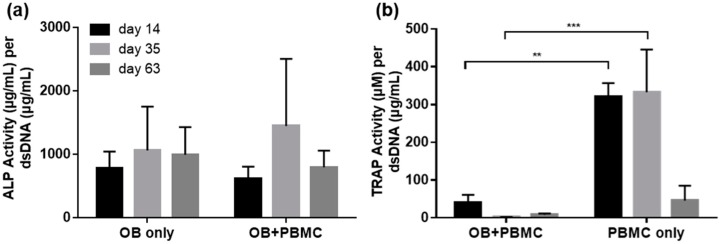
(**a**) Alkaline phosphatase (ALP) and (**b**) tartrate-resistant acid phosphatase (TRAP) activities of cells seeded on CaP-coated PCL scaffolds (mean ± SEM, N = 4 with n = 3 for each biological repeat). ** *p* < 0.05 and *** *p* < 0.01.

**Figure 5 ijms-20-01068-f005:**
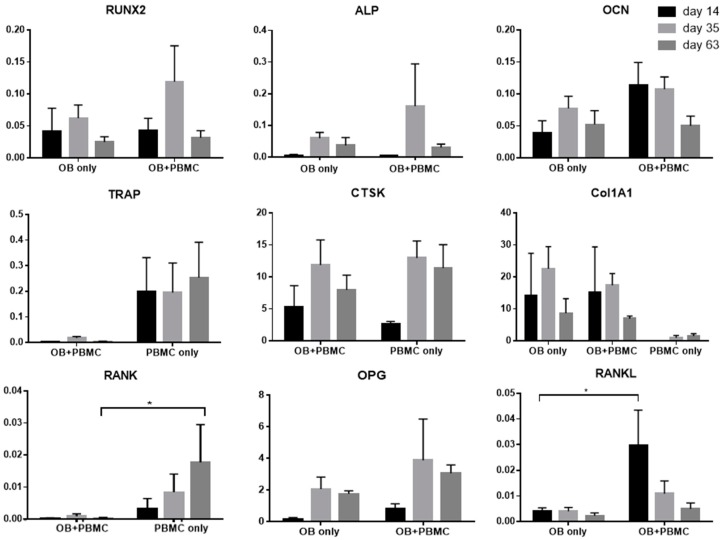
Gene expression (ΔCT method) relative to β-tubulin (house-keeping gene). Mean ± SEM, N = 4. * *p* < 0.05.

**Table 1 ijms-20-01068-t001:** Sequence and characteristics of primers used for qPCR.

Gene	Sequence	Tm (°C)
*RunX2*	Forward: TGCCTAGGCGCATTTCAGGTGC Reverse: TGAGGTGACTGGCGGGGTGT	60
*ALP*	Forward: ACGTGGCTAAGAATGTCATC Reverse: CTGGTAGGCGATGTCCTTA	60
*OCN*	Forward: CCAGCGGTGCAGAGTCCAGC Reverse: GACACCCTAGACCGGGCCGT	60
*OPG*	Forward: GGGACCACAATGAACAAGCTG Reverse: TGTTTTAGGGAGGTGCCAGG	56
*CTSK*	Forward: GGCCCGAGTGGGACCTGTCT Reverse: CCCSCTGCCSSSSCCGCSTGG	56
*TRAP*	Forward: CCCTCGGAGAAACTGCATCAT Reverse: CATGTCCATCCAGGGGGAGA	60
*RANK*	Forward: TGCCTTGCAGGCTACTTCTC Reverse: CCTGCTGACCAAAGTTTGCC	56
*RANKL*	Forward: GGGCCAGGTTGTCTGCAGCGT Reverse: ACCATGAGCCATCCACCACCAGG	58
*Col1A1*	Forward: GCCAAGAAGAAGACATCCCAGC Reverse: TCCCTTGGCACCATCCAA	58
*β-Tubulin*	Forward: GAGGGCGAGGACGAGGCTTA Reverse: TCTAACAGAGGCAAAACTGAGCACC	60

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
