# Peer review of "A Growth Factor-Free Co-Culture System of Osteoblasts and Peripheral Blood Mononuclear Cells for the Evaluation of the Osteogenesis Potential of Melt-Electrowritten Polycaprolactone Scaffolds"

_ijms, 2019, doi:10.3390/ijms20051068_

Reviewer 1 Report

The paper entitled "A growth factors-free osteoblasts and peripheral blood mononuclear cells co-culture system for the evaluation of osteogenesis potential of melt-electrowritten polycaprolactone scaffolds" presents a study of osteogenic differentiation of two types of cells (OB and PBMC) grown on a scaffold composed by PCL with a layer of calcium phosphate. The manuscript is well organized in the division into chapters and subchapters, nevertheless in some paragraph (like in "Analysis of gene expression") appear confused in presenting results. Although the study is intriguing, the data are neither convincing nor statistically significant in many cases (like in qPCR or ALP figures). Therefore, the manuscript could be improved by further analyses on osteogenic differentiation, such as:

- Western Blot or immunofluorescence analysis on some osteogenic proteins (Col1A1, Osteocalcin, Osteonectin, ALP) produced by cells grown on PCL+CaP; these results could validate qPCr results;

- gene expression analysis with qRT-PCR on cell cycle genes (such as PCNA, Cyclin B1) could ameliorate proliferation results that are presented only with Quant-1 PicoGreen reagent;

- an immunofluorescence analysis on FAK/pFAK proteins is recommended in order to better visualize adhesion of OB and PBMC on PCL/CaP scaffolds.

I will reconsider the manucript after major revision.

Author Response

The paper entitled "A growth factors-free osteoblasts and peripheral blood mononuclear cells co-culture system for the evaluation of osteogenesis potential of melt-electrowritten polycaprolactone scaffolds" presents a study of osteogenic differentiation of two types of cells (OB and PBMC) grown on a scaffold composed by PCL with a layer of calcium phosphate. The manuscript is well organized in the division into chapters and subchapters, nevertheless in some paragraph (like in "Analysis of gene expression") appear confused in presenting results. Although the study is intriguing, the data are neither convincing nor statistically significant in many cases (like in qPCR or ALP figures). Therefore, the manuscript could be improved by further analyses on osteogenic differentiation, such as:

Question (1): Western Blot or immunofluorescence analysis on some osteogenic proteins (Col1A1, Osteocalcin, Osteonectin, ALP) produced by cells grown on PCL+CaP; these results could validate qPCr results;

Answer: We would like to thank the reviewer for the recommendation. As mentioned by the reviewer, all assays (i.e. ALP colorimetric assay; osteogenic-related gene expression, i.e. RunX2, ALP, OCN, from qPCR results; and matrix mineralization by von Kossa staining) showed no significance between the OB and OB+PBMS groups. The purpose of this study is to directly compare the effectiveness of OB-only monoculture and OB+PBMC co-culture in an unstimulated, growth factor-free culture system for the in vitro evaluation of PCL/CaP scaffolds bone regeneration potential. Across all assays, we did not observe any differences in osteogenic differentiation modulated by PCL/CaP scaffolds cultured with OB or OB+PBMC. Hence, performing more in depth osteogenic protein analysis with western blot or immunofluorescence analysis was determined to be out of the scope of this study.

In addition, from the qualitative observation of digital microscopy images (figure 3), the simultaneous presence of OB and PBMC may have more effect on the cell proliferation rate. In the presence of PBMC, instead of osteogenic differentiation and matrix mineralization, it seems like the cell proliferation or ECM secretion mechanism was facilitated. Hence, it might be interesting to look into the underlying biological mechanism for cell proliferation or ECM secretion in OB+PBMC culture compared to OB only culture. However, this is also currently out of the scope of this study and shall be followed up in a separate future study. 

We have added additional text in the Discussion section to clarify this aspect of the study.

Question (2):  Gene expression analysis with qRT-PCR on cell cycle genes (such as PCNA, Cyclin B1) could ameliorate proliferation results that are presented only with Quant-1 PicoGreen reagent;

Answer: We do agree with the reviewer that gene expression analysis with qPCR for cell cycle genes could ameliorate proliferation results obtained from PicoGreen assay. However, in this study, we are focusing on the osteogenic differentiation and mineralized matrix formation aspects for the evaluation of the effectiveness of OB or OB+PBMCs culture system for in vitro evaluation of scaffolds bone regeneration potential. Hence, with the limited cDNA material that we were able to obtaine from our cell-scaffold constructs,  the assessment for osteogenic- and osteoclastics-related genes were prioritized. Indeed, as suggested by the reviewer, it will be highly interesting to look into cell cycle genes to ameliorate proliferation results that are presented only with PicoGreen assay. However, the investigation of the underlying biological mechanism for cell proliferation in OB+PBMC culture compared to OB only culture is currently out of the scope of this study. We highly appreciate the reviewer comment and will take this into considerations in subsequent studies. We do realise that the aim/ focus of the study may not have been conveyed clearly in the manuscript. Hence, we have modified the abstract and the introduction of the manuscript. Please refer to highlighted text in the revised manuscript.

Question (3):  An immunofluorescence analysis on FAK/pFAK proteins is recommended in order to better visualize adhesion of OB and PBMC on PCL/CaP scaffolds.

Answer: We highly appreciate the reviewer recommendation on immunofluorescence analysis on FAK/pFAK proteins to better visualize adhesion of OB and PBMC on PCL/CaP scaffolds. As previously mentioned, this study focuses on the osteogenic differentiation and mineralized matrix formation aspects for the evaluation of the effectiveness of OB or OB+PBMCs culture system for in vitro evaluation of scaffolds bone regeneration potential. Although, we do agree with the reviewer that immunofluorescence analysis of the FAK/pFAK protein will give a better visualization of cells adhesion on the MEW scaffolds, the analysis of cell adhesion on the scaffolds is beyond the scope of this study. Nonetheless, this is indeed a valuable recommendation for our future study.

Reviewer 2 Report

Although this manuscript is an interesting that the effect of co-culture with OB and PBMC on bone formation in vitro, there are some serious problems. There are the major concerns need to be carefully addressed.

 Major comments

1) It is unclear what aim the present study is. If the present experiments firstly examined the effect of the scaffold on co-culture system with osteoblasts and PBMC in absence of cytokine the authors strongly should state the new methods and new findings obtained by the present data. 

2) In PCR, the authors should check the mRNA expressions in early phase of culture, such as 5, 7 10, 14 days etc.

 3) In co-culture system, it is unclear what molecules facilitate the bone formation including osteoblast differentiation. If possible, the authors should examine the direct or indirect promotion of the co-culture system on bone formation using transwell culture etc.

 4) It is unknown that mPCL scaffold with OB+PBMC facilitated the bone formation because o difference in von Kossa between OB only and OB+PBMC, but not ECM production.  If you clarify the difference, you should examine the transplantation experiments in vivo using calvaria defect model animal.

 5) Although CaP-coating mPCL scaffold improved osteoblast differentiation and bone formation, you should add the data in only the scaffold as negative control in all experiments.

Author Response

Question (1): It is unclear what aim the present study is. If the present experiments firstly examined the effect of the scaffold on co-culture system with osteoblasts and PBMC in absence of cytokine the authors strongly should state the new methods and new findings obtained by the present data. 

Answer: We thank you the reviewer for the comment. To clarify, the aim of this study is to directly compare the effectiveness of OB-only monoculture and OB+PBMC co-culture in an unstimulated, growth factor-free culture system for the in vitro evaluation of PCL/CaP scaffolds bone regeneration potential, with the focus on the osteogenic differentiation and mineralized matrix formation aspects.

We have made changes to the abstract and introduction of the paper to clarify the aim of this study.

Please refer to highlighted text in the revised manuscript.

Question (2): In PCR, the authors should check the mRNA expressions in early phase of culture, such as 5, 7, 10, 14 days etc.

Answer: We thank you the reviewer for the comment. However, we do have a slightly different perspective for the time-points for mRNA expression on scaffold culture. Scaffolds provide much larger surface area for cell growth compared to 2D culture. Hence, the time progression of cell growth to cell differentiation to matrix mineralization on scaffolds would be extended compared to 2D culture. In addition, cells of different status, i.e. differentiated cells on scaffolds’ surfaces, proliferating cells across the pores of scaffolds, etc., would concurrently be present across the entire scaffolds construct. The intention of the qRT-PCR was to evaluate the osteogenic differentiation of OBs in the presence/ absence of PBMCs. It is widely known that osteogenic differentiation occurs at highly confluence environment [1]. Hence, based on the light microscopy images observation (as shown in Figure 3), mRNA expression was evaluated in the later-phase of culture (i.e. 14, 35 and 63 days) –80% confluence cells on scaffolds with OB or OB + PBMC.

We have added additional text into Section 2.5 Analysis of Gene expression (result) to justify the choice of time-points for gene expression. Please refer to highlighted text in the revised manuscript.

Question (3):  In co-culture system, it is unclear what molecules facilitate the bone formation including osteoblast differentiation. If possible, the authors should examine the direct or indirect promotion of the co-culture system on bone formation using transwell culture etc.

Answer: We thank the reviewer for the comment. Indeed, it would be highly interesting to further investigate the secreted molecules composition in OB, OB+PBMC, and PBMC culture system. However, the collection of results, i.e. ALP assay, qPCR, von Kossa, in this study showed no differences in terms of osteogenic differentiation between OB and OB+PBMC groups. Instead, it seems like the cell proliferation or ECM secretion mechanism of OB was facilitated in the presence of PBMC. Hence, it might be interesting to further investigate the molecules that facilitates cell proliferative and ECM secretion mechanism. However, this is beyond the scope of this manuscript and has been planned into our future studies. We do hope for the understanding of the reviewer. We regret that this was not conveyed clearly in the manuscript. Hence, we have added new text to the discussion section. Please refer to highlighted text in the revised manuscript.

Question (4): It is unknown that mPCL scaffold with OB+PBMC facilitated the bone formation because of difference in von Kossa between OB only and OB+PBMC, but not ECM production.  If you clarify the difference, you should examine the transplantation experiments in vivo using calvaria defect model animal.

Answer: We highly appreciate the reviewer feedback. We do agree that an in vivo study may offer the potential to better understand the underlying mechanism for bone regeneration guided by MEW PCL/CaP scaffolds. However, this is currently beyond the scope of this manuscript. The intention of the study was to potentially offer an co-culture model for the in vitro evaluation of osteogenic potential of scaffolds intended for bone regeneration application – a system the better mimic the physiological microenvironment of the regenerating bone niche. The aim of the study have not been conveyed clearly in the manuscript. Hence, we have make changes to the abstract and introduction of the paper to clarify the aim of this study. Please refer to highlighted text in the revised manuscript.

Question (5): Although CaP-coating mPCL scaffold improved osteoblast differentiation and bone formation, you should add the data in only the scaffold as negative control in all experiments.

Answer: We thank you the reviewer for the constructive feedback. We have added additional results of PCL scaffolds (without CaP-coating) cultured with OB, PBMC or OB+PBMC to figure 2 (confocal scanning microscopy for cells morphology) and figure 3 (von Kossa staining for matrix mineralization). Based on these qualitative outcome, it is clear that the PCL scaffolds have less osteogenic capability compared to PCL/CaP scaffolds, which was also demonstrated in other studies [2, 3]. In addition, it is apparent that the biomaterial properties of scaffolds (PCL vs. PCL/CaP) affect cell behavior. Hence, for the subsequent experiments, i.e. RT-qPCR, ALP and TRAP assays, PCL/CaP scaffolds were cultured with OB (control), PBMC (control) or OB+PBMC to enable direct comparison of the effectiveness of OB monoculture or OB+PBMC co-culture system for the in vitro evaluation of scaffolds bone regeneration potential. The use of OB and PBMC as control group was stated in line 251-252.

We have added additional text into Section 4: Methods and Materials for the justification and clarifications of experimental groups used across the different assays. Please refer to highlighted text in the revised manuscript.

 [1]       Abo-Aziza, F.A.M. and Z. A A,The Impact of Confluence on Bone Marrow Mesenchymal Stem (BMMSC) Proliferation and Osteogenic Differentiation, International journal of hematology-oncology and stem cell research, 2017. 11(2): p. 121-132.

[2]       Vaquette, C., S. Ivanovski, S.M. Hamlet, et al.,Effect of culture conditions and calcium phosphate coating on ectopic bone formation, Biomaterials, 2013. 34(22): p. 5538-51.

[3]       Poh, P.S.P., D.W. Hutmacher, B.M. Holzapfel, et al.,In vitro and in vivo bone formation potential of surface calcium phosphate-coated polycaprolactone and polycaprolactone/bioactive glass composite scaffolds, Acta Biomaterialia, 2016. 30: p. 319-333.

Round  2

Reviewer 1 Report

The paper is ready for publication.

Reviewer 2 Report

The revised manuscript almost improved in point-out parts.

I recommend the manuscript for publication.